

# Including autapomorphies is important for paleontological tip-dating with clocklike data, but not with non-clock data

Nicholas J. Matzke[1,2] and Randall B. Irmis[3,4]

[1] Division of Ecology and Evolution, Research School of Biology, The Australian National University, Canberra, Australian Capital Territory, Australia
[2] School of Biological Sciences, University of Auckland, Auckland, New Zealand
[3] Department of Geology & Geophysics, University of Utah, Salt Lake City, UT, United States of America
[4] Natural History Museum of Utah, Salt Lake City, UT, United States of America

## ABSTRACT

Tip-dating, where fossils are included as dated terminal taxa in Bayesian dating inference, is an increasingly popular method. Data for these studies often come from morphological character matrices originally developed for non-dated, and usually parsimony, analyses. In parsimony, only shared derived characters (synapomorphies) provide grouping information, so many character matrices have an ascertainment bias: they omit autapomorphies (unique derived character states), which are considered uninformative. There has been no study of the effect of this ascertainment bias in tip-dating, but autapomorphies can be informative in model-based inference. We expected that excluding autapomorphies would shorten the morphological branchlengths of terminal branches, and thus bias downwards the time branchlengths inferred in tip-dating. We tested for this effect using a matrix for Carboniferous-Permian eureptiles where all autapomorphies had been deliberately coded. Surprisingly, date estimates are virtually unchanged when autapomorphies are excluded, although we find large changes in morphological rate estimates and small effects on topological and dating confidence. We hypothesized that the puzzling lack of effect on dating was caused by the non-clock nature of the eureptile data. We confirm this explanation by simulating strict clock and non-clock datasets, showing that autapomorphy exclusion biases dating only for the clocklike case. A theoretical solution to ascertainment bias is computing the ascertainment bias correction ($Mk_{parsinf}$), but we explore this correction in detail, and show that it is computationally impractical for typical datasets with many character states and taxa. Therefore we recommend that palaeontologists collect autapomorphies whenever possible when assembling character matrices.

Corresponding author
Nicholas J. Matzke,
nick.matzke@anu.edu.au

# INTRODUCTION

In parsimony phylogenetic analyses, the only data informative for reconstructing the tree topology are those with grouping information: potentially shared, derived character states (synapomorphies; *Hennig, Davis & Zangerl, 1999*). An autapomorphy—a state unique to one terminal taxon or Operational Taxonomic Unit (OTU; *Mishler, 2005*)—contributes one step to any possible topology. Therefore, autapomorphies are routinely excluded from further analysis in cladistics programs (e.g., the TNT *xinact* and *info* commands (*Goloboff, Farris & Nixon, 2008*); the PAUP* *exclude* command (*Swofford, 2003*); and see *Yeates, 1992*), and autapomorphic characters are often not even collected during assembly of a character-taxon matrix.

In model-based inference, autapomorphies *can* be informative (*Lewis, 2001*; *Wright & Hillis, 2014*), because autapomorphies contribute information about the overall rate of change in the character matrix and site-specific rate heterogeneity. An insufficiently recognized point is that autapomorphies might be particularly important in "tip-dating" analyses, where terminal taxa include fossils with ages older than the present day (*Alexandrou et al., 2013*; *Pyron, 2011*; *Ronquist et al., 2012*; *Wood et al., 2013*). Tip-dating analyses might be expected to be particularly sensitive to autapomorphies: all autapomorphies occur on terminal branches by definition, so their exclusion will shorten the morphological branchlengths of terminal branches (and thus presumably their time branchlengths), and perhaps increase estimated branch-wise rate variation.

An alternative to inclusion of autapomorphies is ascertainment-bias correction, where the likelihood of unobservable character patterns, $L_{unobs}$, is calculated, and the likelihood of the observed data is normalized by dividing by $1 - L_{unobs}$ (*Felsenstein, 1992*; *Lewis, 2001*). The two common corrections are the Markov-$k$ model with an ascertainment bias correction for the unobservability of invariant characters (M$k$-variable-only, or M$k$v; *Lewis, 2001*), and Markov-$k$ with an ascertainment bias correction for parsimony-uninformative characters, M$k_{parsinf}$ (*Allman, Holder & Rhodes, 2010*; *Ronquist & Huelsenbeck, 2003*). These corrections are options in Mr Bayes and can be implemented in Beast1/Beast2 XML, but several studies briefly mention that the scalability and correctness of M$k_{parsinf}$ computations may be problematic (*Dos Reis, Donoghue & Yang, 2016*; *Koch & Holder, 2012*; *Matzke, 2016*).

The effect of inclusion/exclusion of autapomorphies and ascertainment-bias correction has not been studied in a tip-dating context. Datasets appropriate for doing so are rare because they need to systematically collect all autapomorphies, as well as dates for the OTUs. *Müller & Reisz (2006)* constructed an all-fossil, morphological matrix of early eureptiles and tested the effect of inclusion/exclusion of autapomorphies in undated Bayesian inference, and recommended including autapomorphies. *Lee & Palci (2015)* discussed the importance of autapomorphies for tip-dating, but did not test the effect of their inclusion/exclusion. We obtained dates for Müller and Reisz's taxa, and used the dataset to test the effects of autapomorphy inclusion. Surprisingly, no effect on dates was found. This might be due to the non-clocklike nature of the dataset, an explanation we confirm with a simulation study that shows autapomorphy exclusion biases terminal branchlength estimates when

the data are highly clocklike, but not in a non-clock dataset. We also examine the $Mk_{parsinf}$ correction and show that it scales poorly for characters with more than two states, limiting its usability.

## METHODS

### Data

The morphological matrix was taken from *Müller & Reisz (2006)*. The date ranges for OTUs were derived from the literature, following best practices guidelines (*Parham et al., 2012*). Correlation between time and morphological branchlengths in a TNT parsimony analysis was used as a rough assessment of clocklike behavior (for further description of all methods, as well as all data and scripts used, see Supplemental Information).

### Tip-dating eureptiles

Tip-dating in Beast2 (*Bouckaert et al., 2014*; *Drummond & Bouckaert, 2015*) with Birth-Death-Serial Sampling (BDSS) or SA-BDSS (Sampled Ancestors) tree models (*Gavryushkina et al., 2015*; *Gavryushkina et al., 2014*) requires a specialized XML input file. To set this up, we used BEASTmasteR (*Alexandrou et al., 2013*; *Matzke, 2015*; *Matzke & Wright, 2016*), a set of R functions that convert NEXUS character matrices, an Excel file containing tip date ranges, and other priors and settings, into XML. Three different site models were used: $Mk$, $Mkv$, and $Mk_{parsinf}$. The summary Maximum Clade Credibility (MCC) trees were plotted with 95% highest posterior densities (HPDs) on inferred node (blue) and tip dates (red) using BEASTmasteR functions and custom R scripts. Mean node dates, node 95% HPD widths, posterior probabilities, and rates were compared between pairs of analyses (with/without autapomorphies) for nodes/bipartitions shared between analyses ($n = 14$), with the Wilcoxon signed-rank test (WSRT) for paired samples. Due to the small number of tests, no multiple-test correction was used.

### Simulation

To test whether clocklike behavior is needed to observe effects of autapomorphy exclusion on date estimates, a BDSS tree similar in size to the empirical dataset (30 OTUs) was simulated using *TreeSim* (*Stadler, 2015*). A "strict clock" dataset of 1,000 binary characters was simulated on this tree under the $Mk$ model with a rate low enough (0.05) that a substantial proportion of the characters (577/1,000) were invariant or autapomorphic. A "non-clock" dataset was produced by reshuffling the time-branchlengths of the simulated tree, and then simulating another 1,000 characters at the same rate. Datasets were filtered to produce variable-only and parsimony-informative-only datasets, effectively imposing ascertainment bias. Beast2 runs were conducted on both simulated datasets under $Mk$, $Mkv$, and $Mk_{parsinf}$ using the same setup as for the empirical analysis. All scripts, Beast2 inputs and outputs, and further details of the analyses are available in Supplemental Information.

### Scalability of the $Mk_{parsinf}$ correction

Although listed as an option in MrBayes for a over a decade, surprisingly, $Mk_{parsinf}$ has not been formally described anywhere in the literature, leading to widespread lack of knowledge

of how it works and whether or not it is computationally feasible on typical datasets. Nor has there been any formal treatment of its computational scalability. The key issue is the number of unobservable character patterns for a character with a particular number of states, as the likelihood of each unobservable pattern must be calculated. While this is feasible for a binary character (which appears to be the assumption made by MrBayes), for a dataset with many taxa and multistate characters, the number of unobservable site patterns rapidly climbs into the millions. The Appendix contains a derivation of the number of likelihood calculations required by $Mk_{parsinf}$, and a discussion of computational scalability.

## RESULTS

### Tip-dating eureptiles
Fourteen bipartitions were shared by the summary trees of all analyses. MCC trees for two runs are illustrated in Fig. 1; for all runs, see Fig. S1. Summary statistics of key parameters are shown in Table 1. Linear regression of tip age against the root-to-tip distance in a parsimony analysis (the number of morphological steps on all branches leading to a tip, see a similar approach for molecular data by *Rambaut et al., 2016*) indicated that time and parsimony branchlengths were not correlated. This is evidence that the morphological characters in the eureptile dataset are not evolving in a clocklike manner.

### Inferred node dates
Estimates of the root age are almost identical between analyses with and without autapomorphies (Table 1). Comparing mean dates for nodes shared across the MCC trees yields no significant differences (WSRT, two-sided, $n = 14$ shared nodes), with $P = 0.359$ for the $Mk$ inference, and $P = 0.280$ for $Mkv$ inferences.

### Dating uncertainty (HPD widths)
Adding data should reduce uncertainty, especially with small morphological datasets. The null hypothesis, that the no-autapomorphies dataset does not have greater HPD widths, was rejected for the $Mk$ inferences (including vs. excluding autapomorphies, 9.20 vs. 9.94, $P = 0.023$, one-sided WSRT); the result for the $Mkv$ inferences was only suggestive (9.37 vs. 9.66, $P = 0.105$).

### Posterior probabilities (PPs)
PPs were higher for runs including autapomorphies under both the $Mk$ model (including vs. excluding autapomorphies, 0.902 vs. 0.756) and the $Mkv$ model (0.900 vs. 0.835). The null hypothesis, that the no-autapomorphies dataset does not have smaller PPs, was rejected at a significance level of 0.05 for both the $Mk$ inference ($P = 0.0095$, one-sided WSRT) and $Mkv$ inference ($P = 0.0252$).

### Relaxed clock
The mean of the relaxed clock rate is dramatically affected by inclusion of autapomorphies, under both the $Mk$ model (with autapomorphies, rate mean = 0.0782 changes per site per million years, 95% HPD [0.015–0.159]; without: 0.788 [0.0305, 3.982]) and the $Mkv$

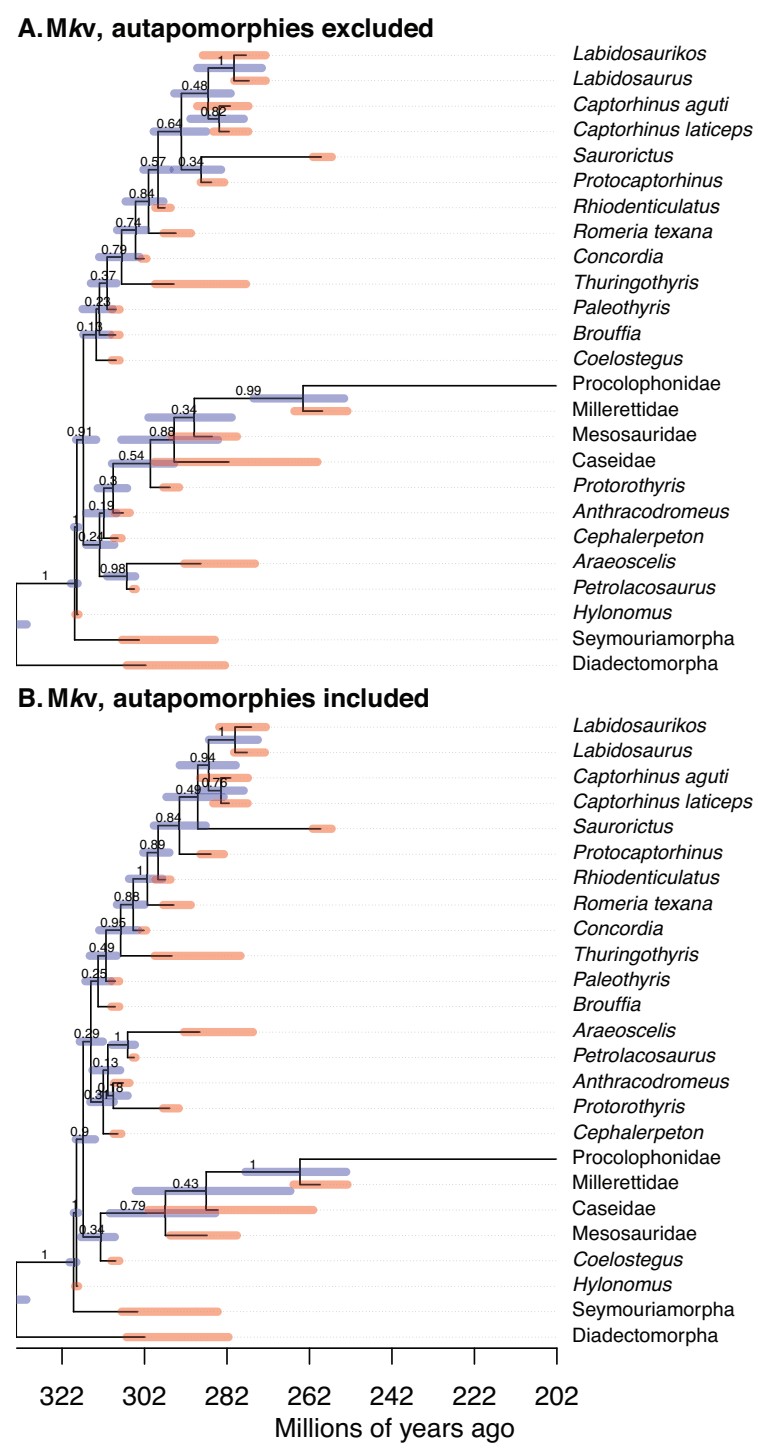

**A. M*kv*, autapomorphies excluded**

**B. M*kv*, autapomorphies included**

Millions of years ago

**Figure 1** **Comparison of the tip-dated phylogenies of early eureptiles inferred when excluding (A) or including (B) autapomorphies, under M*kv* ascertainment bias correction.** Numbers are posterior probabilities. Bars represent the 95% HPD.

**Table 1** Comparison of summary statistics from the five Beast2 runs using "best-practices" tip dates.

| Run # | 1 | 2 | 3 | 4 | 5 |
|---|---|---|---|---|---|
| Data | Including autapomorphies | | Excluding autapomorphies | | |
| Model | M$k$ | M$k$v | M$k$ | M$k$v | M$k$-parsinf |
| Ln posterior | −1393.4 | −1362.2 | −1154.2 | −1144.9 | −1134.4 |
| ESS | 1,801 | 1,485 | 1,801 | 1,801 | 1,801 |
| Root age | 332.6 [330.2, 335.3] | 332.5 [330.0, 335.1] | 332.6 [330.1, 335.1] | 332.6 [330.1, 335.1] | 332.6 [330.0, 335.1] |
| Birth | 0.360 [0.0355, 1.316] | 0.424 [0.0405, 1.708] | 0.342 [0.0463, 1.221] | 0.381 [0.0402, 1.377] | 0.564 [0.0444, 2.841] |
| Death | 0.336 [9.17e−5, 1.315] | 0.3995 [1.13e−4, 1.723] | 0.318 [4.97e−6, 1.220] | 0.357 [2.57e−4, 1.391] | 0.541 [6.37e−4, 2.843] |
| Sampling | 0.0271 [7.90e−4, 0.0626] | 0.0264 [0.00104, 0.0650] | 0.0271 [8.85e−4, 0.063] | 0.0261 [9.96e−4, 0.0634] | 0.0256 [7.66e−4, 0.0643] |
| Clock rate mean | 0.0782 [0.015, 0.159] | 0.0376 [0.0074, 0.0840] | 0.788 [0.0305, 3.982] | 0.550 [0.0228, 2.655] | 0.235 [0.0142, 0.664] |
| Clock rate SD | 1.747 [1.201, 2.399] | 1.712 [1.111, 2.309] | 2.436 [1.572, 3.477] | 2.341 [1.488, 3.379] | 2.079 [1.318, 2.984] |

model (with: 0.0376 [0.0074, 0.0840]; without: 0.550 [0.0228, 2.655]) (tests in Supplemental Information), roughly a increase of an order of magnitude in both cases. The M$k_{parsinf}$ run of the no-autapomorphies dataset yielded an intermediate clock rate (0.235, 95% HPD [0.0142–0.664]).

## Simulations

Figure 2 shows the simulation procedure and key comparisons. Similar tree topologies were inferred under all datasets, but estimated time-branchlengths differed. When the characters are clocklike and autapomorphies are included, inferred time-branchlengths are highly accurate (Fig. 2B). However, when autapomorphies are excluded, inferred terminal branchlengths are biased downwards, and accuracy decreases for all branchlengths. The effect in Fig. 2C can also be seen by comparing inference while including vs. excluding autapomorphies, when the characters are clocklike (Fig. 2D), but this effect disappears for non-clock data (Fig. 2E).

## Feasibility of M$k_{parsinf}$

Equations in the Appendix demonstrate that M$k_{parsinf}$ can be feasible for 2-state characters, and for 3-state characters on small datasets (~10 times slower for our dataset), but rapidly becomes computationally impractical as the number of taxa or states increases. The number of unobservable site patterns for various combinations of numbers of taxa and character states are shown in Table 2.

## DISCUSSION

Although estimated mean rate parameters for the eureptile dataset dropped dramatically (by 10 times or more) when autapomorphies were included (and somewhat less when ascertainment-bias correction was used instead), the downstream effects on confidence

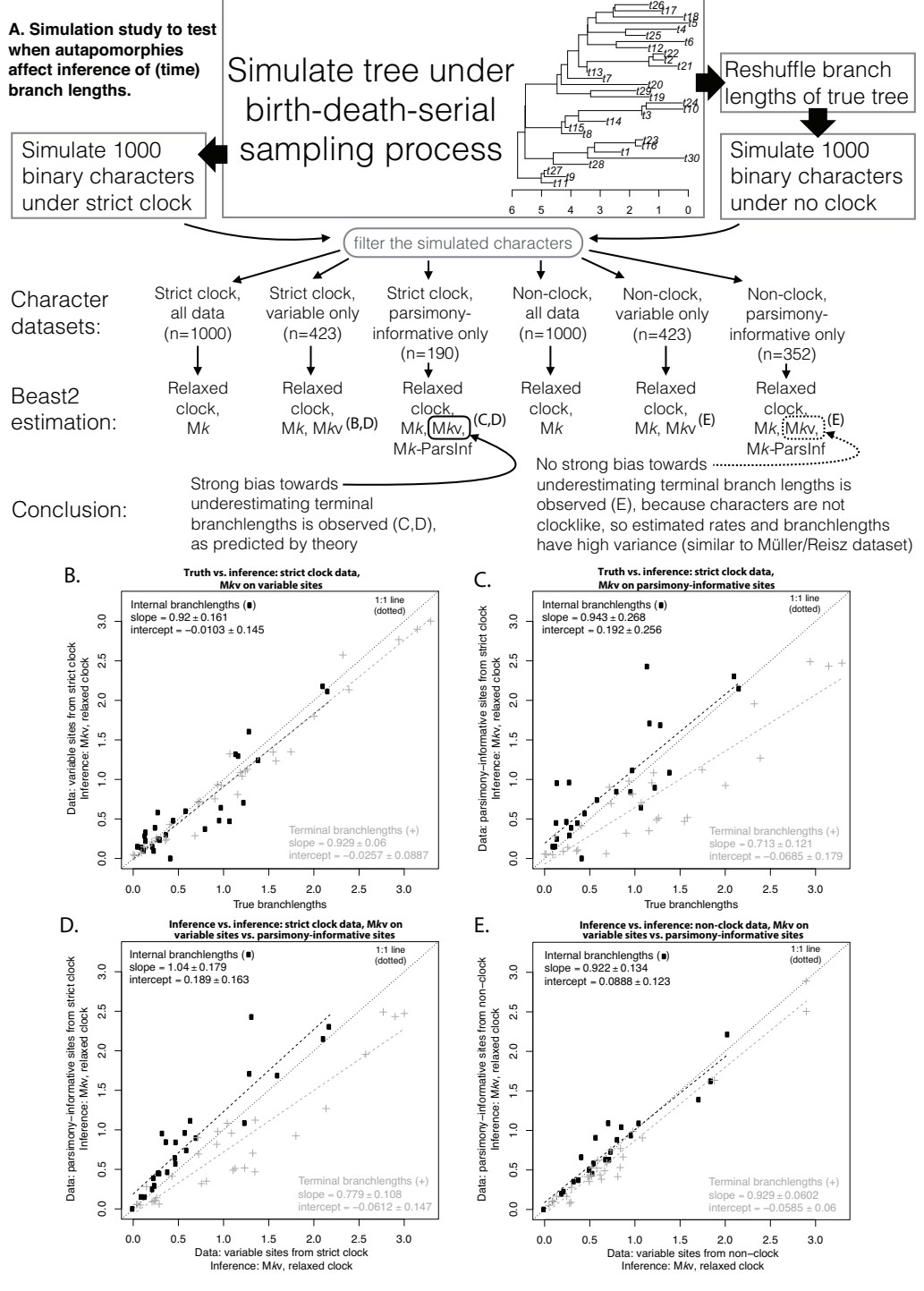

**Figure 2** **Simulation procedure and results.** Simulation procedure (A) and results (B–E). The lack of an effect of excluding autapomorphies on dating in the empirical eureptile result is similar to the result on non-clock data shown in (E).

**Table 2  Number of patterns that are unobservable in the $Mk_{parsinf}$ model.**

|  |  | # states: 2 | 3 | 4 | 5 | 6 |
|---|---|---|---|---|---|---|
|  | 4 | 10 | 63 | 292 | 1,045 | 3,006 |
|  | 5 | 12 | 93 | 544 | 2,505 | 9,276 |
|  | 10 | 22 | 333 | 4,084 | 42,505 | 381,546 |
|  | 20 | 42 | 1,263 | 32,164 | 730,005 | 15,085,086 |
| # of taxa | 50 | 102 | 7,653 | 500,404 | 30,062,505 | 1,698,527,706 |
|  | 100 | 202 | 30,303 | 4,000,804 | 490,250,005 | 57,089,105,406 |
|  | 200 | 402 | 120,603 | 32,001,604 | 7,921,000,005 | 1.87E+12 |
|  | 500 | 1,002 | 751,503 | 500,004,004 | 3.11E+11 | 1.86E+14 |
|  | 1,000 | 2,002 | 3,003,003 | 4,000,008,004 | 4.99E+12 | 5.97E+15 |

were small (Table 1; Supplemental Information), and there was no detectable effect on date inference. This seems surprising, because the exclusion of autapomorphies must reduce the number of morphological changes on terminal branches. However, reflection on the interaction between non-clocklike data, and the flexibility of relaxed-clock Bayesian tip-dating methods, provides an explanation. If the character data are non-clocklike, then the method will estimate a high rate of branchwise rate variation, indicating lack of correlation between time elapsed and morphological branchlength. In this situation, most of the dating information for the analysis comes from the serial-sampling of fossil tips rather than morphological branchlengths. If morphological branchlength is not correlated with time, this remains true whether or not autapomorphies are included, and adding autapomorphies is not likely to change the dating inference.

Our simulation results (Fig. 2) confirm this explanation. The analysis of the empirical eureptile dataset is likely similar to the situation shown in Fig. 2E: inferred time branchlengths are roughly the same whether or not autapomorphies are included. However, on a clocklike dataset, exclusion of autapomorphies clearly has an effect (Fig. 2C). This suggests that the importance of including autapomorphies in tip-dating analyses depends on whether or not the characters have clocklike behavior. Unfortunately, assessing clocklike behavior will be more difficult when autapomorphies have been ignored or gathered only inconsistently (as is common).

An alternative to coding autapomorphies is the $Mk_{parsinf}$ model. However, the Appendix shows that it scales too poorly to be generally useful for characters with large number of states (Table 2; Supplemental Information). All versions of MrBayes back to at least 3.1.2 allow a "coding=informative" ascertainment bias correction to be specified, but the increase in computation time for a run with a single discrete character is very similar whether the character has 2, 3, 4, or 5 states (tested on MrBayes versions 3.1.2 through 3.2.6, and the 3.2.7 development version). This suggests that $Mk_{parsinf}$ may be implemented assuming only binary characters, and may be formally incorrect for multistate characters (as briefly noted by *Dos Reis, Donoghue & Yang, 2016*; *Matzke, 2016*), despite many usages in the literature. However, as most morphological datasets are dominated by binary characters, this issue may have limited impact on inference, and requires further study.

## CONCLUSION

Our study indicates that the common practice of repurposing character matrices devised for parsimony and undated Bayesian analyses may not be sufficient in the world of Bayesian tip-dating. For higher quality datasets (many characters, clocklike behavior), the bias in dating introduced by ignoring autapomorphies may become significant. Additionally, ascertainment bias corrections are at present computationally impractical for many datasets with multistate characters. Finally, autapomorphies have additional utility for improving estimates of rates and rate variation, for species identification, for measuring disparity, and because autapomorphies may become synapomorphies when new taxa are described. Therefore, we recommend that autapomorphies be coded and used whenever possible.

## ACKNOWLEDGEMENTS

We would like to thank April Wright (ORCID ID: 0000-0003-4692-3225), David Bapst (ORCID ID: 0000-0002-9087-1103), Adam Huttenlocker, Graeme Lloyd (ORCID ID: 0000-0001-6887-3981), Daniel Ksepka, James Parham (ORCID ID: 0000-0002-5221-0072), Nathan Smith, Alan Turner, and Mike Lee for helpful discussions, and especially helpful comments from Mark Holder (ORCID ID: 0000-0001-5575-0536) and two other reviewers.

## APPENDIX: DERIVATION OF THE M$k_{parsinf}$ ASCERTAINMENT BIAS CORRECTION, AND PROBLEMS WITH SCALABILITY

One potential alternative to our recommendation to code autapomorphies could be to employ the M$k$-Parsimony-Informative model (M$k_{parsinf}$), that is, the Markov-$k$ model with an ascertainment bias correction for not just the unobservability of invariant characters, but also for the unobservability of autapomorphic characters (in a dataset that excludes autapomorphic characters).

A significant question is whether or not the M$k_{parsinf}$ model can actually be employed on typical datasets. MrBayes, since at least version 3.1.2, does allow the M$k_{parsinf}$ ascertainment bias correction as an option ("lset coding = informative"; *Ronquist, Huelsenbeck & Teslenko, 2011*, p. 146, or http://mrbayes.sourceforge.net/wiki/index.php/Evolutionary_Models_Implemented_in_MrBayes_3#Standard_Discrete_.28Morphology.29_Model), but does not contain an extensive description of how it works, and the model does not seem to have been formally described in any publication. *Allman, Holder & Rhodes (2010)* analyse model identifiability in the M$k_{parsinf}$ context, but only cite *Nylander et al. (2004)* for the model; Nylander et al. in turn cite Ronquist and Huelsenbeck, "in prep.," which appears to be an uncorrected reference to their citation of the MrBayes 3 publication (*Ronquist & Huelsenbeck, 2003*).

There is therefore a need to explore M$k_{parsinf}$ in detail. Ascertainment bias correction works by enumerating site patterns that are unobservable, calculating their likelihood under the model, and then re-normalizing the observed data likelihood at each site by dividing

by $1 - L$, where $L$ is the likelihood of the unobservable site pattern(s). In *Felsenstein (1992)*, the unobservable pattern was "no restriction site observed," i.e., a column of all 0s. In the M$k$v model, the unobservable patterns include a column of all 0s, a column of all 1s, etc., up to the number of character states, $n$, in a particular character.

However, the situation becomes much more complicated for the M$k_{\mathrm{parsinf}}$ ascertainment bias correction (Table S1). The difficulty (mentioned briefly in *Koch & Holder, 2012*; *Dos Reis, Donoghue & Yang, 2016*; *Matzke, 2016*) is that the number of unobservable site patterns scales very poorly with number of character states and number of taxa. For example, for a 100 taxon data matrix and a 3-state character, the following is an unobservable site pattern: a column that consists of all 0s, a single 1 for taxon 99, and a single 2 for taxon 100. But any other variant of this pattern is also unobservable: all 0s, except state 1 at position 1, and state 2 at position 2, etc. Additional unobservable patterns include all 1s except for two taxa and all 2s except for two taxa. Also unobservable are all patterns that are invariant (all 0s, all 1s, all 2s), and all patterns that are invariant except for one taxon.

Formally speaking, if $n$ is the number of taxa, and $k$ is the number of states in a character, then there are $k^n$ possible patterns for that character. We can calculate the number of site patterns that are unobservable under M$k_{\mathrm{parsinf}}$ by first selecting the number of character states, $i$, found in a particular unobservable site pattern. For example, in a character assumed to have $k = 5$ states, the number of states found in a particular pattern could be $i = 1$ (i.e., an invariant site), $i = 2, \ldots, i = 5$. For each $i$, there are

$$\binom{k}{i} \tag{1}$$

ways to pick which of the $k$ character states will be found in found in the site pattern. Conditional on $i$ character states in a particular unobservable site pattern, one of them will be the "dominant" state (held by all taxa, except for the autapomorphic taxa), and $(i - 1)$ of the character states will be autapomorphies. There are

$$\binom{i}{1} \tag{2}$$

ways to choose which character state is dominant. Conditional on the dominant character state, there are

$$\binom{n}{i-1} \tag{3}$$

ways to choose which taxa will be autapomorphic. Conditional on which taxa are autapomorphic, there are $_{i-1}P_{i-1}$ permutations of ways to assign the $(i - 1)$ character states to the $(i - 1)$ autapomorphic taxa. This is calculated

$$\frac{(i-1)!}{((i-1)-(i-1))!} = \frac{(i-1)!}{0!} = (i-1)! \tag{4}$$

Taking the product of Eqs. (1)–(4) and summing over all $i$ yields

$$\sum_{i=1}^{k} \binom{k}{i}\binom{i}{1}\binom{n}{i-1}(i-1)!$$

$$\sum_{i=1}^{k} \binom{k}{i} i! \binom{n}{i-1} \qquad (5)$$

autapomorphic patterns that are unobservable under the $Mk_{\text{parsinf}}$ ascertainment bias correction, for a character with $k$ states. This equation is implemented in the R function *num_unobservable_patterns_ParsInf*, included in the Supplemental Information (and available online via GitHub Gist, at: https://gist.github.com/nmatzke/ 8f80723b6e1fc80ed5ac).

Calculating the number of unobservable patterns for a range of numbers of taxa and states (Table S1) shows that, for a 100-taxon morphological matrix, the presence of characters with 3 states in the matrix will necessitate calculating the likelihood for 30,303 additional site patterns. This is computationally imaginable, although it will substantially slow the MCMC search for a morphological dataset, which usually has only a few hundred characters. The presence of a 4-state character requires 4,000,804 unobservable patterns. For a 6-state character there are over 57 billion.

In Beast2, unobservable site patterns have to be physically listed in the XML input; even with a script to write out the patterns, users can certainly imagine the difficulty of saving and manipulating XML files containing millions of unobservable patterns. Inspection of the MrBayes code seems to indicate that the $Mk_{\text{parsinf}}$ correction assumes binary characters only (which is computationally feasible; Table S1); but this leaves open the question of what calculation is being done on characters with more than two states.

This is problematic, as many researchers (e.g., *Dembo et al., 2016*) are probably under the impression that $Mk_{\text{parsinf}}$ ascertainment bias correction works for any number of character states. It is possible that this issue is of little significance. After all, most morphological characters are binary. Also, as the number of taxa and character states increases, the fraction of the total number of possible patterns ($k^n$) that are unobservable (Eq. (5)) decreases precipitously (Tables S2 and S3). Thus, perhaps the likelihood of unobservable sites dwindles to irrelevance. This seems to be the observation made in the MrBayes manual (*Ronquist, Huelsenbeck & Teslenko, 2011*, pp. 146–147), where the authors state they observed that as the total tree length (sum of branchlengths in terms of number of expected changes per site) increases e.g., above 20–30 taxa, the ascertainment bias correction becomes negligible. However, this may depend greatly on the "true" rates—if they are low, and thus invariant and autapomorphic patterns are some of the most probable patterns, then the likelihood correction from unobservable patterns could be large. As this paper showed, in the case of the 25-taxon eureptile dataset, just switching from $Mk$ to $Mkv$ models dropped the mean clock rate estimate by about 1/3 in both the autapomorphies-included and autapomorphies-excluded dataset. This would affect the morphological branchlengths (number of expected changes per site) in a similar way. As $Mk_{\text{parsinf}}$ includes the $Mkv$ correction, this suggests $Mk_{\text{parsinf}}$ would have the same or greater effect.

Two comments we received from readers of a draft of this manuscript deserve attention. First, M Lee, pers. comm., 2016 pointed out that the equations above would be somewhat different if the researchers coding characters excluded not only parsimony-uninformative characters, but also characters that were "partially uninformative." An example would be

the character pattern 00112—character states 0 and 1 are potential synapomorphies, but character state 2 is an autapomorphy. Above, we have focused on the "literal" interpretation of "parsimony-informative," which we think is the understanding commonly used in the literature and in programs. We suggest that an ascertainment bias correction that assumes the unobservability of invariant, parsimony-uninformative, and partially-parsimony-informative characters should have a new name, perhaps simply "partial-parsinf."

Second, M Holder, pers. comm., 2016 pointed out that the scalability problem is less detrimental, although still daunting, if it is realized that some patterns will have the same likelihood under the M$k$ model (because it is a symmetric-equal-rates model). For example, the patterns 00112, 00221, 11002, 11220, 22110, and 22001, will all have the same likelihood. Therefore the log-likelihood can be calculated for one of these patterns, and multiplied by the number of patterns in that category. This amounts to removing $i!$ from Eq. (5), and using it as a weight to multiply by the log-likelihood of a pattern. Beast2 does have a "weight" option for its Alignment class, but we have not tested it in combination with the ascertained/excludefrom/excludeto options in the XML.

Equation (5) applies to unordered characters, where any autapomorphies will be parsimony-uninformative. If it is instead assumed that the characters are ordered, then any pattern with more than two states will be parsimony-informative. For example, the pattern 011112 would be parsimony-uninformative for an unordered character, but parsimony-informative for an ordered character, because bipartitions grouping states $(0, 1)$ and $(1, 2)$ would be favoured over trees grouping $(0, 2)$. Thus, the number of unobservable patterns (assuming the researchers doing the character scoring had this in mind when building their matrix) is much reduced, since only patterns with 1 or 2 character states are unobservable. The equation is:

$$\sum_{i=1}^{2} \binom{k}{i}\binom{i}{1}\binom{n}{i-1}(i-1)! \tag{6}$$

The unobservable pattern counts for an ordered character are shown in Table S4, and fractions in Table S5.

Resolution of the discussion about when and where M$k_{\text{parsinf}}$ is functional, useful, or unnecessary may be difficult, as it depends in part on gnarly philosophical questions about what the "complete" morphology matrix would look like (how many invariant morphological characters are "truly" observable in any particular clade?). This is closely tied to another difficult question: what is the "true" morphological rate for "all" of the morphology? We can briefly suggest that probably such questions are almost unanswerable in the abstract, and that any meaningful statements about rates and completeness must be made with reference to some method of character collection. It certainly appears that these problems should be studied more carefully than can be done here. Unless these issues are resolved, however, it may be that including all codeable autapomorphies, and using the M$k$v ascertainment bias correction, is the best option.

### Funding

Nicholas Matzke was supported by Discovery Early Career Researcher Award (DECRA) DE150101773, funded by the Australian Research Council, and by The Australian National University. He was also supported by the National Institute for Mathematical and Biological Synthesis (NIMBioS), an Institute sponsored by the National Science Foundation, the US Department of Homeland Security, and the US Department of Agriculture through NSF Awards #EFJ0832858 and DBI-1300426, with additional support from The University of Tennessee, Knoxville. Randall B. Irmis was supported by NESCent and the University of Utah. The funders had no role in study design, data collection and analysis, decision to publish, or preparation of the manuscript.

### Grant Disclosures

The following grant information was disclosed by the authors:
Discovery Early Career Researcher Award (DECRA): DE150101773.
National Institute for Mathematical and Biological Synthesis (NIMBioS).
Institute sponsored by the National Science Foundation.
US Department of Homeland Security.
US Department of Agriculture through NSF: EFJ0832858, DBI-1300426.
The University of Tennessee, Knoxville.
NESCent.
The University of Utah.

### Competing Interests

The authors declare there are no competing interests.

### Author Contributions

- Nicholas J. Matzke conceived and designed the experiments, performed the experiments, analyzed the data, prepared figures and/or tables, authored or reviewed drafts of the paper, approved the final draft.
- Randall B. Irmis conceived and designed the experiments, contributed reagents/materials/analysis tools, authored or reviewed drafts of the paper, approved the final draft.

### Data Availability

Matzke N, Irmis RB. Data from: Tip-dating studies should include autapomorphies: an example with eureptiles. Dryad Digital Repository. https://doi.org/10.5061/dryad.8q4c8.

### Supplemental Information

Supplemental information for this article can be found online at http://dx.doi.org/10.7717/peerj.4553#supplemental-information.

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
