# Peer review of "Including autapomorphies is important for paleontological tip-dating with clocklike data, but not with non-clock data"

_PeerJ, doi:10.7717/peerj.4553_

## Round 0.1 · original submission · Minor Revisions

Dear authors,

I have received highly positive comments on your manuscript from three reviewers. All suggest that this is an important study that should be published following few, minor revisions. I agree with the comments of the reviewers. I think this is a well-written and carefully presented study that will have important implications for future research that uses tip-dating. I am happy to accept the paper following attention to the minor comments (particularly of Reviewer #2 in the pdf attachment).

Minor editorial points:

Line 47: should be “—“ not “--"
Line 50: delete “even”
Line 67: “MyBayes” has been italicised elsewhere in the ms, please check for consistent use throughout
Line 72: should be “The effect of..”
Line 75: delete duplicated reference
Line 79: add “(2006)”
Line 87: delete “Mueller and Reisz”
Line 88: reference missing before “2012”
Line 118: should be “for over a decade”
Line 147: “but not for the Mkv inferences” rather than “suggestive”
Line 167: should be “shown in Figure 2b”
Line 192: If starting a new paragraph here, I would make a specific statement here rather than write “this explanation”
Line 203: check use of italics for “MrBayes”
Line 215: replace “in the world of” with “for”
Line 247: “Felsenstein” should not be in parentheses

Reviewer 1 ·

Basic reporting

See General Comments Below

Experimental design

See General Comments Below

Validity of the findings

See General Comments Below

Additional comments

This is a fine, succinct discussion of an important issue (whether morphological characters should be "filtered" before phylogenetic analysis). It should be publishable in PeerJ after the following minor comments are addressed.

OTUs - Recommend inserting (=terminal taxa) on first use of term OTU.

>These corrections are options in MrBayes and can be implemented in Beast2 XML
Can also be done in BEAST 1.

>Linear regression of tip age against the root-to-tip distance in a parsimony analysis (the number of morphological steps on all branches leading to a tip) indicated that time and parsimony branchlengths were not correlated.

This approach has been used before, e.g. in Path-o-gen / Tempest by Rambaut, which probably would be good to cite here.

>tested on MrBayes versions 3.1.2 through 3.2.6
Might be fair to point out that “coding=informative” is now no longer listed in the command manual (though the code is still there) - perhaps in recognition of potential issues.

Useful reference to cite: "Why remove autapomorphies?" by Yeates in Cladistics 1992

·

Basic reporting

The paper is well-written (I've picked up a few minor typos) and the analyses are well-documented in the Main Text and extensive supplementary materials.

Experimental design

The paper is restricted to a single "real world" dataset, plus simulated datasets - to what extent other "real datasets" would behave similarly is an interesting, open question. However, I think the detailed analyses of the eureptile dataset are sufficient to cover the issues of concern.

Validity of the findings

As above, it would be interesting to see how other datasets would behave, but the results presented are certainly valid for the datasets being analysed.

Additional comments

I have made quite a few comments/suggested changes on the marked up pdfs (main text and supplementary info) that I would like the authors to address, but these are relatively minor. Otherwise, I think this is an interesting, timely paper that will be of broad interest to systematists, palaeontologists etc.

·

Basic reporting

No Comments

Experimental design

No Comments

Validity of the findings

No Comments

Additional comments

This is a brief, but important and consequential paper on the behavior of various tip-dating methods in the presence of autapomorphies. It is very important to demonstrate the effect on estimated clock-rates and relative lack thereof for estimated dates. Similarly evaluating the Mk_parsinf correction is a necessary advancement for researchers using these techniques. I have essentially no revisions needed for publication.

---

## Round 0.2 · accepted · Accept

Thank you for addressing all minor points raised by the reviewers, and for responding to all points raised in the annotated pdf. Given that the reviewers praised the quality of this manuscript and that the minor suggestions have been fully incorporated into this new version, I think the paper is now ready to be published. I look forward to seeing the paper published!